# A Short Report on the Polymerization of Pyrrole and Its Copolymers by Sonochemical Synthesis of Fluorescent Carbon Dots

**DOI:** 10.3390/polym11081240

**Published:** 2019-07-26

**Authors:** Moorthy Maruthapandi, Aharon Gedanken

**Affiliations:** Bar-Ilan Institute for Nanotechnology and Advanced Materials, Department of Chemistry, Bar-Ilan University, Ramat-Gan 52900, Israel

**Keywords:** carbon dots, sonochemical method, UV-light, polypyrrole, Poly(pyrrole-co-aniline), Poly (Bis(p-aminophenyl) ether-co-pyrrole)

## Abstract

In polymer chemistry, polymerization constitutes the process of the conversion of monomers into polymers using an initiator to form polymeric chains. There are many polymerization techniques and different systems exist by which the polymers are classified. Recently, our group has reported the synthesis of polymers using both carbon dots (CDs) and UV light as initiators. In these reports, the carbon dots were used with or without UV light. The CDs produce free radicals in the presence of UV light, indicating their role as initiators. The CD surface has many unshared or unpaired electrons, making it negatively charged. The present study focuses on the use of CDs for the formation of polymers from monomers containing various functional group. The properties of the synthesized CDs and the polymers obtained from the various monomers were characterized by various analytical techniques, including Fourier-Transform Infrared (FTIR) spectroscopy, X-ray Diffraction (XRD), Thermogravimetric Analysis (TGA) and Solid-State NMR spectroscopy. This polymerization technique is of interest both from the scientific aspect and for its applicative potential. The synthesized polymers were investigated for their various applications.

## 1. Introduction

Recently, carbon dots (CDs) have been emerging as new materials demonstrating great potential for use in biomaterial and medical engineering. In the last few decades, carbon dots have begun to feature in such areas as electronic industries, solar and photovoltaic cells, aerospace, catalysis, targeting drug delivery [1,2,3,4,5]. CDs are a type of nanomaterial composed of a carbon core, with oxygen and hydrogen on the circumference of the 5 nm dots appearing as carboxylic, or carbonyl, or alcohol groups [6,7,8,9,10]. CDs have been used as new cell imaging and drug-delivery nano-platforms due to their outstanding fluorescence, biocompatibility, excellent aqueous solubility, negligible cytotoxicity, easy functionalization, biodegradation ability, and environmental friendliness. CDs have been synthesized by several methods, including microwaves and ultrasonic irradiation [11,12,13,14,15,16,17]. The characteristic properties of CDs are partially dependent on the particular carbon-based starting materials, which include sugars, proteins, amino acids, juice carbohydrates, milk, glucose, etc. [12,18,19,20]. The CDs synthesized sonochemically from polyethylene glycol (PEG-400) provide a high quantum yield (QY) of fluorescence, [21,22] and have been applied in numerous areas of science and technology due their high photoluminescence, biocompatibility, low-cost preparation, no cell cytotoxicity, and high stable fluorescence [5,21,23,24,25,26,27,28,29]. Moreover, the synthesis of CDs from PEG produces a good passivating layer on their surface. These sonochemically prepared CDs were used here as an initiator for polymerization because they form free radicals in aqueous solution, which leads to their unique functionality and their high potential in many applications [1,6,7,8,30,31,32]. The free radicals play a main role in the polymerization mechanism [18,19,20,33]. Sonochemistry usually yields unorganized and perturbed particles having dislocations and vacancies leading to the formation of free radicals in water. This was demonstrated, for example for metal oxide NPs comparing the formation of commercial and sonochemically made products of the same size [34]. The same happens in the current work, where the CDs prepared sonochemically has yielded OH radicals. The radicals are detected by EPR when exposed to spin traps [22]. Polypyrrole (PPY) is recognized as a conducting polymer that relies on its electrochemical responsive properties rather than on its chemical structure. PPY offers a potentially useful resource because it possesses optical, magnetic, and electronic properties comparable to those of metals or semiconductors, while also retaining its polymeric structure and properties, such as ease of processing, low toxicity, flexibility, and adjustable electrical conductivity [34,35,36,37,38,39,40]. However, the general applications of PPY have been mostly limited to date owing to its poor mechanical properties. In order to improve the mechanical properties, various dopants have been introduced into the polymer using certain strong initiators [39,41,42,43]. Our novel polymerization of PPY was developed using only CDs as initiator, without any other strong initiators [44,45,46]. The synthesized polypyrrole has been used as an adsorbent material for various organic dyes and for biological application [47,48]. 

To date, there have been only a few reports on the synthesis of polymers using CDs and UV light, all of which were reported by our group. The conventional synthesis of the conductive polymer polypyrrole usually employed various strong oxidizing agents as initiators. The best methods by which PPY is polymerized are either through chemical oxidization or electrochemically. There are many initiators that have been used for the polymerization of pyrrole, including ferric chloride, silver nitrate, ammonium persulfate, potassium persulfate, and copper (II) chloride [49,50,51,52,53,54,55,56,57,58,59,60]. These oxidizing agents, however, dangerous as they can cause eye, nose, throat, lung, and skin irritation upon contact [56]. The persulfate salts also produce asthmatic symptoms. We developed polymerization techniques for the synthesis of PPY and PPY copolymers using the CDs both with and without UV-light. Copolymers of PPY were also prepared from various monomers containing various functional group using CDs as initiators. The CD surface has many unshared or unpaired electrons, making it negatively charged [61,62,63]. During the course of the reaction, the positively charged pyrrole is attracted to the carbon dots, prior to the polymerization reaction. The reflux method was used to synthesize the copolymer of polypyrrole within 24 h without UV light [64]. The UV light promoted free radical generation, replacing the use of thermal heating to accomplish the polymerization. The present article reports on the synthesis of PPY and its copolymers using CDs, as well as discussing the coating of the CD onto a glass slide that served as a substrate for the polymerization of PPY. 

## 2. Experimental Section

### 2.1. Preparation of CDs

30 mL of PEG-400 was placed in a 50 mL beaker in an oil bath and heated to 70 °C. The tip of an ultrasonic transducer was immersed in the PEG-400 and sonication was carried out for 3 h at 65% amplitude [21,22].

### 2.2. CDs Coated on Glass Slides

In parallel, CDs were produced and coated on glass slides. The process was identical to that described above except that the sonication beaker contained a glass slide and sonication was continued for 60 min at 25 °C in order to deposit the preformed CDs onto the glass slide. The glass slides were washed with water and dried in an oven at 50 °C for 1 h under ambient atmosphere. The free CDs observed on the slides were utilized for the synthesis of the polymer. The ultrasonic waves, and particularly the micro-jets formed after the collapse of the acoustic bubbles, caused the embedding of the CDs formed on the glass [44]. After the synthesis, the polymer was washed several times with water and ethanol to remove unreacted CDs. The lifetime of the CDs containing free electrons is short and their quantity is much smaller than the amount of the polymer. Sonochemically prepared CDs initiated polymerization is displays in Scheme 1.

### 2.3. Synthesis of Polypyrrole and Its Copolymer by CDs

Pyrrole (1.0 g) was mixed in 30 mL of 1 M nitric acid in a 100 mL beaker at room temperature. A 3 mL aqueous solution containing 9 mg of CDs was then added. A similar process was carried out to prepare polyaniline using CDs coated onto a glass slide rather than a CD dispersion in water. To activate the reaction, the solution was kept under UV light (USV-18 EL series UV lamp and 8-Watt, 365 nm wavelength) for 3 days, after which we observed the precipitation of blackish-brown solids, which we collected by filtration. These solids were washed several times with distilled water and dried at room temperature. The polypyrrole was also synthesized using different molar concentrations of nitric acid and pyrrole [44]. The Scheme 2 shows the chemical reaction of polypyrrole. 

### 2.4. Poly(pyrrole-co-aniline)

Pyrrole (1.0 g) and aniline (1.0 g) were mixed in 30 mL of 1 M nitric acid in a 100 mL beaker at room temperature. To this, 5 mL of an aqueous solution containing 15 mg of CDs was added. The polymerization reaction was stimulated by illumination with UV light for three days, after which a blackish-brown solid was obtained and collected by filtration, washed several times with distilled water, and dried at room temperature [44]. The chemical reaction of poly(pyrrole-co-aniline) is illustrates in Scheme 3.

### 2.5. Synthesis of Poly (Bis(p-aminophenyl)ether-co-pyrrole)

Bis(p-aminophenyl) ether (0.6 g, 3 mmol), pyrrole (0.6 g, 10 mmol), CD solution (3 mL, 9 mg), acetonitrile (15 mL), nitric acid (1.5 mL), and distilled water (10 mL) were mixed, after which the color of the monomer changed from a white solid to a purple solution. The reaction mixture was then mixed well and refluxed in an oil bath for 24 h at 90 °C under ambient environment. The reaction mixture was then cooled to room temperature and poured into a beaker filled with ice. The dark brown solids that precipitated were collected by filtration, washed several times with distilled water, and dried at room temperature [64]. The chemical reaction of (Bis(p-amino phenyl) ether-co-pyrrole) is demonstrates in Scheme 4.

## 3. Physical Characterization of CDs as Initiator

Characterization of the properties of the prepared CDs, used as an initiator for the different polymerization processes, is provided below. High-resolution transmission electron microscopy (HR-TEM) images of the CDs revealed them to be spherical in shape, with a rather narrow size distribution (Figure 1a). The average size of the CDs was about ~10 nm, which is in contrast to our previously-prepared CDs. CDs embedded on the glass slide are displayed in Figure 1b. The particles on the glass slide present a larger size range (17–50 nm) than those formed in the solution. This larger size appears to be due to the sonochemical coating bombarding the already-coated surface and adding CDs to the first existing layer and thus enlarging them. The detailed characterizations of XRD, EPR, Zeta potential values, and XPS for CDs were already provided by our group [44].

Recently, we reported the formation of ultrafine CDs by means of ultrasonic cavitation in PEG-400. Characterization of the products was performed employing various methods. HRTEM images (Figure 2a) of the CDs reveal them to possess a spherical shape with a narrow size distribution (~6 nm). The fluorescence emission of the CDs found in the supernatant was spread over the 420–610 nm range (Figure 1b), with the excitation wavelengths being between 330 and 490 nm. The d-spacing of CDs was estimated from the HRTEM and selected area electron diffraction patterns and was found to be 0.21 nm. 

Figure 3a presents images of a CD dispersion taken under ambient light and under irradiation with a hand-held, long-wave TLC lamp (365 nm excitation), highlighting the strong, blue fluorescence of the CD dispersion resulting from such excitation. The UV-Vis absorption spectra of the CDs are displayed in Figure 3b. The strong absorption bands of CDs at 246 and 360 nm are due to the π–π* electron transition and graphitic nature of the carbon material, respectively.

A magnified HRTEM image (Figure 3c, inset) of individual CDs demonstrates that the lattice plane of CDs is 0.21 nm and ties with the graphite facet (100), in agreement with previous reports on CD microscopic images [26]. The fluorescence emission spectra were recorded at various excitation wavelengths (330–470 nm) and maximum emission from such excitations were observed in the range of 420–600 nm. Figure 3d displays the typical fluorescence spectra of CDs produced from PEG-400 following the sonication treatment.

### 3.1. Polypyrrole

The formation of polypyrrole was confirmed by XRD, FT-IR and TGA (Figure 4). The X-ray diffraction of polypyrrole displayed in Figure 4a, a broadband around 2θ = 13°–45° due to being parallel to the polypyrrole structure and perpendicular to the polymer chain. The XRD pattern reveals the synthesized polypyrrole to be amorphous in nature. The FT-IR spectrum of polypyrrole shows in Figure 4b. The FT-IR spectrum reveals a broadband around 3261 cm^−1^ due to N-H stretching vibration. The peak at 1574 cm^−1^ is due to the C=C in-ring stretching vibration and the band at 1130 cm^−1^ is due to the C–N stretching vibration. The bands at 1308 and 1078 cm^−1^ are due to the C–H in-plane vibration. The thermal stability of the synthesized polypyrrole is provided in Figure 4c. The TGA analysis displays three stages of weight loss: the first stage is minor and the second and third are major weight loss [44].

The particle size distribution of the polypyrrole was measured by dynamic light scattering (DLS). Two different populations of particle size distribution were found: 75% of the particles were micron-size, ca. 1000–1600 nm; and 25% were nano-size, ca. 600–900 nm. The SEM image shows particles of 2–7 μm. The difference between the SEM and the DLS results from the large micron size particles in the SEM being due to the aggregation of particles. A careful look at Figure 5b reveals even smaller particles. 

In order to confirm successful polymerization, the ^13^C solid-state NMR was analyzed. The spectrum (Figure 6) is displays four broad bands at 110, 122, 126, 129 ppm and one shoulder peak at 142 ppm. The peaks at 122 ppm and 126 are assigned to carbon C-1 and C-2, respectively, while the peak at 110 ppm originates from protonated C-3 and non-protonated C-7 carbon of the quinoid ring in the polypyrrole chain. The peak around 129 ppm is attributed to the protonated C-6 and C-4 and the non-protonated C-4 carbons.

### 3.2. Poly(pyrrole-co-aniline)

X-ray diffractograms of poly(pyrrole-co-aniline) are given in Figure 7a. A broad peak around 2θ = 18 to 40 degrees, indicating the formation of the repeated unit of both the monomers in the polymer chain, demonstrates that the polymer structure is highly associated. The broad peak is evidence that the formation of poly(pyrrole-co-aniline) is amorphous in nature. The IR spectrum for poly(pyrrole-co-aniline) is provided in Figure 7b. The FT-IR shows a broadband at 3255 cm^−1^, corresponding to the N–H stretching vibration. The aromatic C–H stretching vibration band appears at 2984 cm^−1^ and the peaks at 1572 and 1486 cm^−1^ are attributed to the C=C stretching mode of the benzenoid and quinoid rings. The peak at about 1170 cm^−1^ is due to the C–N stretching vibration and the peak at 1190 cm^−1^ corresponds to the N=Quinoid-N+ stretching vibration, respectively [44].

TGA of the Poly(pyrrole-co-aniline) shows a three-step weight-loss performance in Figure 7c. The first major weight loss of 5% occurs within the temperature range of 70–120 °C and is due to the removal of moisture or loss water from the polymer; the second major weight loss of around 32% occurs at around 212–490 °C and is due to with removal of small oligomers; the third major weight loss is 22% within a range of 455–720 °C, and is due to the decomposition of the polymer. The morphology of the poly(pyrrole-co-aniline) was examined by SEM. Figure 7e shows that poly(pyrrole-co-aniline) is formed as irregular shapes, ca. 400–500 nm, while the 10-micron size particles in the SEM are due to the aggregation of the particles. Confirmation of the poly(pyrrole-co-aniline) product was established using ^13^C solid-state NMR in a cross-polarization magic angle-spinning mode. The ^13^C solid-state NMR spectrum for the poly(pyrrole-co-aniline) is given in Figure 7d. It comprises a broad peak, centered at 127 ppm, and companied by a few peaks that appear as shoulders The spectral features all correspond to the aromatic carbons of the poly(pyrrole-co-aniline) backbone. The spectrum demonstrates six broad resonances which can be observed at 116 ppm (shoulder), 138.0, 146.1, 154.5, and 164.0 ppm. The peak at 164 ppm originates from protonated and non-protonated carbons of the quinoid part of the poly(pyrrole-co-aniline) structure.

### 3.3. Poly (Bis(p-aminophenyl)ether-co-pyrrole)

The FTIR spectrum poly (Bis(p-aminophenyl)ether-co-pyrrole) is displayed in Figure 8a as follows: broad peak at 3351 cm^−1^ is assigned to the N-H stretching vibration; an aromatic C=C–H stretching vibration at 3074 cm^−1^; peaks at 1605 and 1501 cm^−1^, due to the aromatic C=C stretching vibration; peaks around 1340 cm^−1^ which are due to the C–N stretching vibration; a peak at 1246 cm^−1^, is due to the ether C–O stretching vibration; and a peak at 830 cm^−1^, is assigned to the out-of-plane bending vibration of an amine group. A summary of all significant IR peaks and the corresponding structural evidence is given in Table 1 [64].

The TGA weight loss of Bis(p-aminophenyl) ether exhibited three dissimilar weight loss stages (Figure 8b): (1) Minor weight loss (4%) at 97–135 °C, due to the evaporation of moisture and water molecules; (2) 12% weight loss between 468–664 °C, attributed to the thermal removal of oligomers; and (3) final weight loss above 721 °C due to the thermal decomposition of the polymer.

Further demonstration of the crystalline nature of the successful polymer was achieved by XRD analysis, with the XRD spectrum of Poly (Bis(p-aminophenyl) ether-co-pyrrole) given in c. A broad peak around 2θ = 13.4°–26.9° can be ascribed to the repeated units of pyrrole. The XRD spectrum shows that the copolymer is amorphous in nature, as a highly crystalline polymer would display much sharper peaks. A morphological study of the solid-state polymer was conducted using scanning electron microscopy (SEM), with SEM images of Poly (Bis(p-aminophenyl) ether-co-pyrrole) given in Figure 9a. The SEM image of Poly (Bis(p-aminophenyl)ether-co-pyrrole) reveals agglomerated particles with irregular spherical shapes and a broad range of diameters between 800 and 1500 nm.

The particle size distribution of the Poly (Bis(p-aminophenyl)ether-co-pyrrole) was analyzed by dynamic light scattering (Figure 9a). The particle size diameters ranged from 600 to 2010 nm. The DLS analysis revealed the copolymer of Poly (Bis(p-aminophenyl)ether-co-pyrrole) to possess a macro-nano particle size distribution. The polydispersity of the copolymer with dispersity indices of less than 1.0, evidences the particulate nature of the polymers and strongly signifies the good processability of copolymer dispersions [59].

The successful copolymerization was confirmed by ^13^C solid-state NMR. The solid-state NMR spectrum Poly (Bis(p-aminophenyl) ether-co-pyrrole) (Figure 10) displays a peak “a” at 153.3 ppm due to the carbon singly-bound to oxygen: and a peak “b” at 144.1 ppm attributed to the carbon singly-bound to nitrogen. The peak “c” at 136 ppm is due to the carbon, which is part of the C–N bond between the Bis(p-aminophenyl) ether subunit and the pyrrole ring. Peaks “d” and “g”, at 123.5 and 106.3 ppm, are assigned to the aromatic pyrrole carbons. Finally, peaks “e” and “f” indicate that the peaks at 121.1 and 118.3 ppm are attributed to the aromatic carbons of Bis(p-aminophenyl) ether in the copolymer unit.

## 4. Mechanism for Polymerization

The polymerization mechanism is supported by CDs and UV light. The dots have an unshared pair of electrons on their surface and are carry negative charges [21,31,61,63]. The amine group of the aniline is transformed into the protonated form by the nitric acid. The reason for choosing the high concentration of nitric acid (1–4 M) is that it can efficiently react with base, forming Phenyl ammonium ion, (C_6_H_5_NH_3_+) at room temperature, this ion can easily react with CDs. Another argument is that the high concentration of the acid can accelerate both the oxidation and the acidification of the amine group. Both influence thermodynamic and kinetics, favor the high concentration of the acid. Electron negative charges on the CDs are adsorbed to the positively charged of the aniline cations, through electrostatic interactions. The role of the UV light in this reaction is the activation of aniline, which enables more aniline molecules to form the protonated species. This is well known and reported recently in the literature [57]. Carbon dots were shown to produce OH radicals in water. The OH radicals remove one hydrogen, forming H_2_O. A radical NH_2_+ group is left on the aromatic group. In the next stage, a proton is removed when two of these radicals interact leading to the formation of the polymer. Thus the oxidizing agent is the OH radical. The UV light assists in the formation of the OH radicals [22]. The formation of more free radical is caused by the photoexcitation of CDs. The mechanism for the formation of polyaniline by carbon dot is shows in Scheme 5.

## 5. Conclusions

This short report provides in-depth data on the polymerization of pyrrole and its copolymer, using carbon dots as an initiator. The disadvantages and limitations in using various other initiators associated with the polymerization of pyrrole are noted, and how the use of CDs overcomes these problems is highlighted in this report. In addition, we note that both polypyrrole and copolymers were easily synthesized using only CDs as an initiator, making the polymerization low cost, non-toxic, and requiring an easy one-step preparation method. CDs are recognized as an effective and economical initiator at low concentration and can be used for polymer synthesis. The synthesized conductive polypyrrole produced using CDs has also been used in applications such as in adsorbent material for organic dyes due to its recyclable capacity, morphology, and stability. Among the available polymeric adsorbent materials, the CD-initiated polypyrrole and copolymer macro-nano materials have been widely explored as efficient adsorbent materials for the removal of different organic dyes from aqueous sources. This property has been attributed to the presence of highly active surface sites in the macro-nano adsorbents. In order to improve the performance of the CD-initiated polypyrrole, several metal oxides were used to form composites with PPY, especially for biological application. The CD-initiated polypyrrole was composited, for example, with CuO and demonstrated an efficient antibacterial effect against both *E. coli* and *S. aureus*. The present report thus introduces a novel polymerization method and demonstrates its advantages for PPY and its copolymers.

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
