# Peer review of "A Short Report on the Polymerization of Pyrrole and Its Copolymers by Sonochemical Synthesis of Fluorescent Carbon Dots"

_polymers, 2019, doi:10.3390/polym11081240_

Round 1
Reviewer 1 Report
The study described in this paper is very interesting and quite original. As far as I know, reporting CD-induced polymerization of small molecules induced by UV light is a first in the literature. The results are quite convincing and may pave the road for further, useful, developments. Therefore, I have no doubts in recommending publication. I only have two minor concerns that should be addressed before publication.
1) The paper is very elusive in discussing the possible mechanism(s) by which CDs induce the polymerization mechanism, leaving many doubts to the reader. I would suggest the authors to add, at the end of the discussion secion, an entirely new paragraph where the possible microscopic mechanism responsible of their findings is discussed, even in the form of a speculation.
Essentially, the reader of this paper would certainly like to know what exactly is the role of CDs in promoting polymerization, and, in particular, what CD surface groups are involved in the reaction. As another important point, the authors should explicitly clarify whether the CDs take part to the formed polymer network, or only act as catalysts which are then washed away in the last step of the synthesis.
2) Considering that the field of CDs is so rapidly evolving, I have the feeling that the references quoted in the introduction should be updated with more recent papers. In particular, I suggest the authors to add in their references list one or more of the most recent reviews of the field, such as the following two: https://doi.org/10.3390/c4040067 and https://doi.org/10.1039/C8GC02736F
Author Response
Response to the Reviewers
Reviewer: 1
The study described in this paper is very interesting and quite original. As far as I know, reporting CD-induced polymerization of small molecules induced by UV light is a first in the literature. The results are quite convincing and may pave the road for further, useful, developments. Therefore, I have no doubts in recommending publication. I only have two minor concerns that should be addressed before publication.
Question: 1. The paper is very elusive in discussing the possible mechanism(s) by which CDs induce the polymerization mechanism, leaving many doubts to the reader. I would suggest the authors to add, at the end of the discussion section, an entirely new paragraph where the possible microscopic mechanism responsible of their findings is discussed, even in the form of a speculation.
Response: Thank you for the suggestion. The detailed mechanism has been added in the manuscript.
Question: 2. Considering that the field of CDs is so rapidly evolving, I have the feeling that the references quoted in the introduction should be updated with more recent papers. In particular, I suggest the authors to add in their references list one or more of the most recent reviews of the field, such as the following two: https://doi.org/10.3390/c4040067 and https://doi.org/10.1039/C8GC02736F
Response: The suggested references have been cited.
Reviewer 2 Report
The Author presented a short report on the polymerization of pyrrole and 2 its copolymers by sonochemical synthesis of 3 fluorescent carbon dots. The data they have shown for the polymerization of different organic units is quite convincing and worth acceptable. However, the manuscript is lacking to present the facts rather it is more biased to explain the data. I have attached some comment that the author must have done to make their job more impactful and convincing to the audience. It could be accepted after addressing the following queries:
1) According to the author, the CDs prepared by the ultrasonic method are negatively charged which is acting as a polymerization initiator. So to justify its negative charge the author could have shown the zeta potential value.
2) If the author is using Polyethylene glycol as a precursor so how do these carboxylic functionalities are appearing on the CDs surface shown in scheme 1.
3) The author has not mentioned the nature of the CDs. The author must have shown the PXRD data. And try to co-relate the fate of polymerization with the crystallinity and amorphous nature of the CDs.
4) The author has not mentioned anything about the purification after completion of the reflux reaction to segregate the polymer from the CDs. The author is supposed to describe the fate of these CDs after the polymerization which is missing from the manuscript.
5) The authors show the production of carbon dots from their TEM images. But the images do not represent a fine contrast. For instance in Figure 3 (c), the HRTEM image does not show the presence of particles at all. SAED diffraction patterns and hexagonal structure of CDs if shown by the authors would have confirmed and presented a strong evidence for the carbon dots formation.
6) The authors have also not shown P-XRD spectra of the formed CDs.
7) The authors have talked about the presence of OH radicals formation in Line 49 but not shown any supporting data. Why?
8) The authors did not report any purification steps for the synthesized CDs and the synthesized polymers as well, artifacts may have been created.
9) All the FTIR and the NMR data will be unreliable if correct purification or retrieval steps for the polymers from CDs solution has not been performed. Since, the FTIR and NMR spectra of the synthesized products may be observed because of the already present unreacted pyrrole and PEG moieties
Author Response
Reviewer: 2
The Author presented a short report on the polymerization of pyrrole and 2 its copolymers by sonochemical synthesis of 3 fluorescent carbon dots. The data they have shown for the polymerization of different organic units is quite convincing and worth acceptable. However, the manuscript is lacking to present the facts rather it is more biased to explain the data. I have attached some comment that the author must have done to make their job more impactful and convincing to the audience. It could be accepted after addressing the following queries:
Question: 1. According to the author, the CDs prepared by the ultrasonic method are negatively charged which is acting as a polymerization initiator. So to justify its negative charge the author could have shown the zeta potential value.
Response: There have been so many reports that the CDs has negative charge on their surface. The Zeta potential for CDs which is prepared from PEG by ultrasonic method was already published by our group. Please see the reference 22 & 23. We have mentioned the reference to the Zeta Potential values on the bottom of page 4.
Question: 2. If the author is using Polyethylene glycol as a precursor so how do these carboxylic functionalities are appearing on the CDs surface shown in scheme 1.
Response: It is well known that CDs have on their surface have functional groups such as carboxylates, aldehydes and OH groups, independent of their preparation source. In the current case producing the CDs from PEG by sonication the OHs and the carboxylates are originated from the PEG due to the radicals formed by the sonication. These functional groups on the CD surface are mentioned in thousands of papers. Why do we have to repeat this well known fact? Moreover, it is demonstrated in Scheme 1 in our paper.
Question: 3. The author has not mentioned the nature of the CDs. The author must have shown the PXRD data. And try to co-relate the fate of polymerization with the crystallinity and amorphous nature of the CDs.
Response: All the detailed physical and chemical characterizations for CDs from PEG were already published by our group. Please see the references 22 and 23.
Question: 4. The author has not mentioned anything about the purification after completion of the reflux reaction to segregate the polymer from the CDs. The author is supposed to describe the fate of these CDs after the polymerization which is missing from the manuscript.
Response: After the synthesis the polymer was washed several times with water and ethanol to remove unreacted CDs. The life time of the CDs containing free electrons is short and their quantity is much smaller than the amount of the polymer. See page 3 lines 90-93
Question: 5. The authors show the production of carbon dots from their TEM images. But the images do not represent a fine contrast. For instance, in Figure 3 (c), the HRTEM image does not show the presence of particles at all. SAED diffraction patterns and hexagonal structure of CDs if shown by the authors would have confirmed and presented a strong evidence for the carbon dots formation.
Response: The strong evidence for the formation of the CDs by all the morphology characterizations have been already published by our group. Please see the references 22 & 23.
Question: 6. The authors have also not shown P-XRD spectra of the formed CDs.
Response: The P-XRD is shown in our previous publication please see reference 23.
Question: 7. The authors have talked about the presence of OH radical’s formation in Line 49 but not shown any supporting data. Why?
Response: The radical formation of CDs has been monitored by EPR measurements. The EPR qualitatively but not quantitatively reveals the production of OH radicals. Please see reference 23.
Round 2
Reviewer 2 Report
The manuscript could be accepted now as the authors answered all the queries
This manuscript is a resubmission of an earlier submission. The following is a list of the peer review reports and author responses from that submission.
Round 1
Reviewer 1 Report
This review was narrowly focused on the polymerization of pyrrole and the CDs made from sonochemical synthesis as well as just collected the data from the authors’ previous two papers [40, 57]. I do not appreciate the easy way to create a review paper. Other major concerns are given below.
(1) Please explain why the sonochemically prepared CDs formed free radicals in aqueous solution? Why do not the CDs, prepared by other methods, such as hydrothermal and microwave irradiation, form radicals?
(2) How to qualify and quantify the radicals produced by CDs and CDs/UV? The physical characterization of CDs, observed in Fig. 2 and Fig. 3, is not important.
(3) How to avoid the embedding of CDs in the polymer products and to approve the polymer products containing no CDs embedded?
(4) What polymers did the CD radicals initiate besides the PPY and its copolymers?
Author Response
Reviewer: 1
Question: 1. Please explain why the sonochemically prepared CDs formed free radicals in aqueous solution? Why do not the CDs, prepared by other methods, such as hydrothermal and microwave irradiation, form radicals?
Response: The facile one-step sonochemical method to prepare the CDs was developed by our group, and it takes 30 minutes. to synthesize the CDs. The sonochemical synthetic route produces free radicals because the products are highly perturbed and unorganized. The product has vacancies and dislocations. That results in free radicals not only in this case but in the many papers we have published on this topic. See for example our paper "The influence of the crystalline nature of nano-metal oxides on their antibacterial and toxicity properties" Perelshtein, I. ; Lipovsky, A. ; Perkas, N. ; Gedanken, A. ; Moschini, E. ; Mantecca, P. NANO RESEARCH, 8, 695-707 (2015). The paper compares the ability of Commercial synthesized vs. sonochemically synthesized metal oxides of the same size.
Question: 2. How to qualify and quantify the radicals produced by CDs and CDs/UV? The physical characterization of CDs, observed in Fig. 2 and Fig. 3, is not important.
Response: The radical formation of CDs has been checked by EPR measurement. The EPR is qualitatively but not quantitively reveals the production of OH radicals.
Question: 3. How to avoid the embedding of CDs in the polymer products and to approve the polymer products containing no CDs embedded.
Response: After the synthesis the polymer was washed several times with water and ethanol to remove unreacted CDs. The life time of the CDs containing free electrons is short and their quantity is much smaller than the amount of the polymer.
Question: 4. What polymers did the CD radicals initiate besides the PPY and its copolymers?
Response: We have synthesized different polymers containing various functional groups such as poly (4,4 diaminodiphenyl methane), polyaniline, poly (oxybis benzamine) and its copolymers etc.
Reviewer 2 Report
This review summarized the work in Dr. Gedanken's lab using carbon dots and UV light as initiators for the polymerization of various polymers or copolymers. I believe the idea is good. However, I am sorry to say that this manuscript is drafted in a rough way. So I can't recommend this manuscript to be published in the current state. All the reasons are listed below.
1. The abbreviation of carbon dots should be consistent throughout the whole manuscript. Authors used more than 3 forms of carbon dots including "carbon dots", "CDs", "CD", "C-dots". You need to keep them uniform.
2. In the abstract, authors said the properties of CDs and polymers were characterized by conventional materials science methods. However, if the reader is not from this field, it will be difficult for them to understand what you indicate. So authors need to specify them such as "spectroscopic and microscopic methods" or "UV/vis, fluorescence, FTIR, TEM, DLS, XRD..."
3. There are many English grammar and style mistakes. For the former, authors need to distinguish the singular and plural forms and when you should use "the". For the latter, some unit of temperature needs your attention. Also, In line 11 of abstract, "exist" is not the right word there. You need a English native speaker for help check the whole manuscript.
4. In line 25-26 of page 1, photocatalysis is one type of catalysis. You should delete "photocatalysis". In line 26, what do you mean "targeting" here? Drug delivery?
5. In line 27, I suggest authors referring to a paper about the elemental composition, size and surface functional groups of carbon dots. "Y. Zhou, S. Sharma, Z. Peng, R. Leblanc, Polymers in Carbon Dots: A Review, Polymers, 9 (2017) 67-85".
6. We should not say carbon dots can dissolve in water. Instead, water-dispersible is more accurate. Please revise similar problem in the whole manuscript. For example, "solution..."
7. The synthetic approaches of carbon dots should be more than microwave and ultrasonic. You can check the same paper I mentioned previously to find more.
8. In line 36 of page 1, "luminescence" should be changed as photoluminescence to be specified.
9. In line 43, please delete "rather than its chemical structure."
10. In line 44, you should change "related to" to "comparable to".
11. In line 47-49, authors need to write details about "In order to improve the mechanical properties various dopants have been introduced in the polymer by using certain
strong initiators".
12. I don't like the way to extend sentences by adding too many unimportant details. For example, line 45, line 58...
13. In line 63-65, there exists contradiction, "During the course of the reaction the positively charged pyrrole is attracted to the carbon dots. This attraction occurs prior to the polymerization reaction."
14. In line 71, thirty should be changed as 30.
15. In line 79, why free CDs will be kept after washing with water?
16. The figure on the top of page 3 has no caption or introduction.
17. In line 89, 98, and 108, blackish- brown precipitate is suspicious to be aggregated CDs.
18. There is no histogram of HRTEM. How can authors convince readers about the size?
19. If all figures have been published in previous work, you need to ask for the permission to use them and demonstrate in each cation of each figure.
20. In line 138, the term is wrong, It should not be p-p* electron transition. And the assignment of peaks are wrong also without references.
21. The content in line 147 is repeated. "330, 350, 370, 390, 410, 430, 450, and 470 nm" should be deleted.
22. In line 166, "6b" should be "5b".
23. The Figure 5 is not clear. I don't even know what occurs ( green lines and dots) to Figure 5b. I don't like the authors' careless attitude for publication.
24. The peaks of FTIR were assigned without references.
25. NMR of carbon13 is not enough to prove the structure of polymers. You need mass spectroscopic data as well as H NMR.
26. Most figures need high resolution and some data in the figures, for example, the peak data.
27. In line 200, authors should not leave a red dot after revision?
28. The references are a little out of date. Please change to newer ones.
Based on my review, I can't recommend this paper to be published. Also, I think authors need a correct attitude towards research and publication. However, I appreciate the idea and would like to check again after the manuscript is revised.
Author Response
Reviewer: 2
Question: 1. 1. The abbreviation of carbon dots should be consistent throughout the whole manuscript. Authors used more than 3 forms of carbon dots including "carbon dots", "CDs", "CD", "C-dots". You need to keep them uniform.
Response: The abbreviation of carbon dots has been arranged in a uniform way.
Question: 2. In the abstract, authors said the properties of CDs and polymers were characterized by conventional materials science methods. However, if the reader is not from this field, it will be difficult for them to understand what you indicate. So authors need to specify them such as "spectroscopic and microscopic methods" or "UV/vis, fluorescence, FTIR, TEM, DLS, XRD...
Response: The sentence has been corrected and the characterization techniques are now specified.
Question: 3. There are many English grammar and style mistakes. For the former, authors need to distinguish the singular and plural forms and when you should use "the". For the latter, some unit of temperature needs your attention. Also, in line 11 of abstract, "exist" is not the right word there. You need an English native speaker for help check the whole manuscript.
Response: The manuscript has now undergone professional editing by a native English speaker.
Question: 4. In line 25-26 of page 1, photocatalysis is one type of catalysis. You should delete "photocatalysis". In line 26, what do you mean "targeting" here? Drug delivery?
Response: Done.
Question: 5. In line 27, I suggest authors referring to a paper about the elemental composition, size and surface functional groups of carbon dots. "Y. Zhou, S. Sharma, Z. Peng, R. Leblanc, Polymers in Carbon Dots: A Review, Polymers, 9 (2017) 67-85".
Response: The reference has now been provided.
Question: 6. We should not say carbon dots can dissolve in water. Instead, water-dispersible is more accurate. Please revise similar problem in the whole manuscript. For example, "solution.
Response: We have modified the paper so that in all places where CD solution is mentioned it was changed to CD dispersion in water.
Question: 7. The synthetic approaches of carbon dots should be more than microwave and ultrasonic. You can check the same paper I mentioned previously to find more
Response:
The facile one-step sonochemical method to prepare the CDs was developed by our group, moreover its easy way to synthesis the CDs within 30 min, no additional energy required such a temperature compare to all other methods, we have explained in the introduction section, why the CDs prepared by sonochemical method, what is the advantages compared to all other methods.
Question: 8. In line 36 of page 1, "luminescence" should be changed as photoluminescence to be specified.
Response: Done.
Question: 9. In line 43, please delete "rather than its chemical structure.
Response: Done.
Question: 10. In line 44, you should change "related to" to "comparable to".
Response: Done.
Question: 11. In line 47-49, authors need to write details about "In order to improve the mechanical properties various dopants have been introduced in the polymer by using certain
strong initiators".
Response:
The detailed information was already provided in our previous publication. Please see the ref. 46
Question: 12. I don't like the way to extend sentences by adding too many unimportant details. For example, line 45, line 58...
Response:
In line 45 we have outlined the properties of polypyrrole. On the other hand, on line 58 we have explained the properties of the initiators. We have provided the detailed information about the other initiators.
Question: 13. In line 63-65, there exists contradiction, "During the course of the reaction the positively charged pyrrole is attracted to the carbon dots. This attraction occurs prior to the polymerization reaction."
Response:
The sentence explains about the mechanism of the polymerization and the interaction between the monomer and initiator. For more information, please see ref. 42.
Question: 14. In line 71, thirty should be changed as 30.
Response: Done.
Question: 15. In line 79, why free CDs will be kept after washing with water?
Response: The Free CDs are involving the polymerization reaction. During the sonication process the CDs might be embedded in the glass plates.
Question: 16. The figure on the top of page 3 has no caption or introduction.
Response: Caption has been added.
Question: 17. In line 89, 98, and 108, blackish- brown precipitate is suspicious to be aggregated CDs.
Response: We have identified the blackish-brown precipitate as the polymer and not CD aggregates
Question: 18. There is no histogram of HRTEM. How can authors convince readers about the size?
Response: The size of the particle was mentioned on the figures. All these figures were already published. For more details about size see the ref. 22, and 41-45. The referee should keep in mind that this is a short review article.
Question: 19. If all figures have been published in previous work, you need to ask for the permission to use them and demonstrate in each cation of each figure.
Response: The references were already provided in each figures. All the work was published and developed by our group.
Question: 20. In line 138, the term is wrong, it should not be p-p* electron transition. And the assignment of peaks is wrong also without references.
Response: Please look at the Fig. 3b
200-400 nm is due to the π –π* transition. Please see the link ( http://www.chemistry.iitkgp.ac.in/faculty/SDG/Spectroscopy%20I.pdf)
Question: 21. The content in line 147 is repeated. "330, 350, 370, 390, 410, 430, 450, and 470 nm" should be deleted.
Response: Done.
Question: 22. In line 166, "6b" should be "5b".
Response: Done.
Question: 23. The Figure 5 is not clear. I don't even know what occurs (green lines and dots) to Figure 5b. I don't like the authors' careless attitude for publication.
Response: The green lines were plotted by the SEM instrument itself to measure the single particle size. We did not plot it manually, also we have mentioned the particle size (2-7 µm) in line 165. For more details about SEM image and the size of PPY please see ref. 42. (The referee should try to understand that this is a short review report of our work and he has to go through all our previous publications before review again).
Question: 24. The peaks of FTIR were assigned without references.
Response: The reference has been assigned
Question: 25. NMR of carbon13 is not enough to prove the structure of polymers. You need mass spectroscopic data as well as H NMR.
Response:
We were unable to measure the molecular weight distribution due to the insolubility of polypyrrole.
Question: 26. Most figures need high resolution and some data in the figures, for example, the peak data.
Response: All the figures were already published; we don’t want to modify or change anything in the figures.
Question: 27. In line 200, authors should not leave a red dot after revision?
Response: Color has been removed.
Question: 28. The references are a little out of date. Please change to newer ones.
Response: See the remark above it is a review article therefore it should mention old papers as well if they are related to the topic.
Reviewer 3 Report
Maruthapandi and Gedanken provide a brief report about the use of fluorescent carbon dots on the polymerization of pyrrole and copolymers. In these procedures, sonochemistry is not employed on the polymer synthesis itself, but on the synthesis of the carbon dots. These nanomaterials will then be used as initiators in the presence of thermal heating or UV light. This could be an interesting addition to this special issue, but before it requires significant improvements.
1. The manuscript needs to be proof-read again, both because the English is not very good, and the writing is somewhat confusing.
2. It is not clear what is the mechanism by which carbon dots act as initiators, both under thermal heating and UV light. The authors talk about radicals, but no proof is given.
3. The “Preparation of CDs” section (page 70) does not indicate how the carbon dots were purified.
4. There is any structural information regarding the carbon dots?
5. The authors should indicate what is the polymerization yield of the carbon dot-initiated reactions.
6. The authors should discuss how the carbon dots compare to other initiators used in polymer synthesis.
7. The synthesis and characterization sections for each polymer should be put together.
Author Response
Reviewer: 3
Question: 1. 1. The manuscript needs to be proof-read again, both because the English is not very good, and the writing is somewhat confusing.
Response:
The manuscript has now undergone professional editing by a native English speaker.
Question: 2. It is not clear what is the mechanism by which carbon dots act as initiators, both under thermal heating and UV light. The authors talk about radicals, but no proof is given.
Response:
The radical formation of CDs has been monitored by EPR measurements. The EPR qualitatively but not quantitatively reveals the production of OH radicals. The mechanism for the polymerization by CDs is explained in our previous publication. Please see the reference 32.
Question: 3. The “Preparation of CDs” section (page 70) does not indicate how the carbon dots were purified.
Response:
The synthesized CDs has been used directly for the polymerization, no purification is required.
Question: 4. There is any structural information regarding the carbon dots?
Response: The structural information for CDs was already published by our group. Please see the ref.22
Question: 5. The authors should indicate what is the polymerization yield of the carbon dot-initiated reactions.
Response:
We have achieved more than 90 percentage yield of the polymer and we have investigated the kinetics for the formation of the polymer. Please see the ref. 59.
Question: 6. The authors should discuss how the carbon dots compare to other initiators used in polymer synthesis.
Response:
In our previous publication we have discussed this issue. We have emphasized the advantage and disadvantage of using other initiators compared with the CDs. Please see the ref.42
Question: 7. The synthesis and characterization sections for each polymer should be put together.
Response: Thanks for suggestion, the paper was already arranged according to the scheme of our report.
Round 2
Reviewer 1 Report
It is a pity that authors still do not face the major issue of the review – this review has just collected the data from the authors’ previous two papers [42, 59]. I do not appreciate the easy way to “create” a review paper. Please check the text again.
Reviewer 2 Report
Authors only addressed half of my comments.
In order for your future publication, I will still list the comments you haven't addressed completely.
The abbreviation of carbon dots should be consistent throughout the whole manuscript. Authors kept all the forms as "CDs" or "CD", which is satisfactory. However, after the abbreviation, you have to constantly use the abbreviation instead of carbon dots.
We should not say carbon dots can dissolve in water. Instead, water-dispersible is more accurate. Please revise similar problem in the whole manuscript. For example, "solution" Authors omit some. For example, in line 111, authors mentioned CD solution.
The synthetic approaches of carbon dots should be more than microwave and ultrasonic. Here I mean you need add more methods in line 33 in the introduction to objectively present the fact to public readers. Then you can explain why you only choose the ultrasonication.
My old comments have not been completely performed by authors. For example, authors didn't replace all "luminescence" to "photoluminescence", which could be found line 38. Also, according to my old comments, authors should have deleted "rather than its chemical structure" in line 43. Also, authors were suggested to change "related to" to "comparable to". However, none of them were changed but authors said "done".
In line 78, authors omit the unit of 30.
My old question "why free CDs will be kept after washing with water" was not answered properly. Authors should explain more or check some references.
Blackish- brown precipitate is still suspicious to be aggregated CDs without persuasive proofs to be polymers.
Authors repeatedly explain this is a short review article. However, authors still need to keep in mind that people should be informed of sufficient data such as histogram of TEM since it can indicate the accuracy and distribution of the particle size.
Again, I insist that if all figures have been published in previous work, you need to ask for the permission to use them and demonstrate in each cation of each figure. This is the format of most review papers.
This is the most important comment here. Authors need to use high-resolution pictures throughout the whole manuscript. Otherwise, it will cause a lot of confusion. For example, according to my old comment, the Figure 5 is not clear. I don't even know what occurs (green lines and dots) to Figure 5b. I checked the reference authors based on authors' comments. However, I found the green dots actually are some numbers.
Reviewer 3 Report
The authors have improved the quality of the writing and presentation, as well as answering all my comments. However, I think that the authors should include in the main text these answers.
For example, they should state in section "Preparation of CDs" that CDs were used without purification. They should also include in the text the polymerization yield calculated by them on ref. 59, and also include on the text a similar discussion made by them on ref. 42 regarding the advantages and disadvantages of using CDs as initiators. They should also provide a summary of the structural data obtained for the CDs on ref. 22.